# The Costs of Complications and Unplanned Readmissions after Pancreatoduodenectomy for Pancreatic and Periampullary Tumors: Results from a Single Academic Center

**DOI:** 10.3390/cancers13246271

**Published:** 2021-12-14

**Authors:** Ralph J. A. Linnemann, Bob J. L. Kooijman, Christian S. van der Hilst, Joost Sprakel, Carlijn I. Buis, Schelto Kruijff, Joost M. Klaase

**Affiliations:** 1Department of Hepato-Pancreato-Billiary Surgery and Liver Transplantation, University Medical Center Groningen, University of Groningen, 9713 GZ Groningen, The Netherlands; r.j.a.linnemann@umcg.nl (R.J.A.L.); b.j.l.kooijman@umcg.nl (B.J.L.K.); joost.sprakel@mst.nl (J.S.); c.i.buis@umcg.nl (C.I.B.); 2Department of Strategic Analytics, University Medical Center Groningen, University of Groningen, 9713 GZ Groningen, The Netherlands; c.s.van.der.hilst@umcg.nl; 3Department of Surgical Oncology, University Medical Center Groningen, University of Groningen, 9713 GZ Groningen, The Netherlands; s.kruijff@umcg.nl

**Keywords:** Whipple, PPPD, pancreas, costs, complication, readmission

## Abstract

**Simple Summary:**

Complications lead to unplanned readmissions (UR) and are reported to be associated with a two- to threefold increase in hospital admission costs. Since healthcare costs are increasing worldwide, cost containment is the major challenge for future healthcare. In the literature, there are only a few studies that analysed hospital costs after pancreatoduodenectomy (PD). In this study, we aimed to create an understanding of the costs of complications and UR in patients who underwent a PD.

**Abstract:**

Background/Objectives: Complications after pancreatoduodenectomy (PD) lead to unplanned readmissions (UR), with a two- to threefold increase in admission costs. In this study, we aimed to create an understanding of the costs of complications and UR in this patient group. Furthermore, we aimed to generate a detailed cost overview that can be used to build a theoretical model to calculate the cost efficacy for prehabilitation. Methods: A retrospective cohort analysis was performed using the Dutch Pancreatic Cancer Audit (DPCA) database of patients who underwent a PD at our institute between 2013 and 2017. The total costs of the index hospital admission and UR related to the PD were collected. Results: Of the 160 patients; 35 patients (22%) had an uncomplicated course; 87 patients (54%) had minor complications, and 38 patients (24%) had severe complications. Median costs for an uncomplicated course were EUR 25.682, and for a complicated course, EUR 32.958 (*p* = 0.001). The median costs for minor complications were EUR 30.316, and for major complications, EUR 42.664 (*p* = 0.001). Costs were related to the Comprehensive Complication Index (CCI). The median costs of patients with one or more UR were EUR 41.199. Conclusions: Complications after PD led to a EUR 4.634–EUR 16.982 (18–66%) increase in hospital costs. A UR led to a cost increase of EUR 12.567 (44%). Since hospital costs are directly related to the CCI, reduction in complications will lead to cost-effectiveness.

## 1. Introduction

A pancreatoduodenectomy (PD) is a complex procedure usually performed in cases of malignant tumours of the pancreatic head or surrounding structures such as the duodenum, the ampulla of Vater, or the distal bile ducts. Over the past decades, the mortality of pancreatic resections has been reduced to less than 5% by centralization of care. However, the overall morbidity of pancreatic resections is still high, ranging from 30% to 60% [1,2,3]. The most common complications after pancreatic resection are anastomotic leakage, especially of the pancreatojejunostomy, also known as a postoperative pancreatic fistula (POPF). POPF is a serious and potentially life-threatening complication [4,5]. Other common complications include delayed gastric emptying (DGE), haemorrhage, intra-abdominal abscess, and wound infection [1,2,3].

With increasing age and multimorbidity, patients are also prone to nonsurgical complications such as pneumonia and cardiovascular events. Complications and the often necessary readmissions are both associated with high hospital costs. In the literature, the median costs for patients undergoing a PD without complications are between EUR 17.000 and EUR 30.000 [6,7,8,9] For patients with major complications, median costs can rise to EUR 50.000–EUR 60.000 [6,7]

The Comprehensive Complication Index (CCI) seems directly related to hospital costs [10]. The CCI is a method to measure the overall postoperative morbidity by severity, based on a ranking by the Clavien–Dindo classification [10].

Since healthcare costs are ever-increasing worldwide, one of the major challenges for the future is balancing costs and potential survival gain, taking quality-adjusted life years into consideration. The primary aim of this study was to gain understanding of the impact of complications and unplanned readmissions, considering the CCI score, on perioperative hospital costs in patients who underwent a PD. Furthermore, we aimed to generate a detailed cost overview that can be used to build a theoretical model to calculate the cost efficacy for a multimodal prehabilitation program [11].

## 2. Materials and Methods

Since July 2013, all patients in the Netherlands who underwent a pancreatic resection have been registered prospectively in the Dutch Pancreatic Cancer Audit (DPCA). This study is a retrospective cohort analysis using the DPCA database of patients who underwent a pancreatoduodenectomy at our tertiary referral center, the University Medical Centre Groningen (UMCG), between 2013 and 2017. In total, 160 patients were included. Pre-, intra- and postoperative variables were obtained. All consecutive adult patients who underwent PD (classic Whipple procedure/pylorus-resecting pancreatoduodenectomy (PRPD) or pylorus-preserving pancreatoduodenectomy (PPPD)) between 2013 and 2017 at our institute were included in the study.

All treatment-related incurred healthcare costs of the index hospital admission and unplanned readmission (UR) related to the PD, from one day before PD until the date of discharge, were obtained from the financial department. These healthcare costs included all components of the surgical procedure, postoperative in-hospital care, postoperative hospital visits within 90 days after discharge, and in-hospital rehabilitation program, and concerned the actual individual patient-related costs that our institute incurred to treat the specific patient for PD. Therefore, a prolonged stay at the ICU or a longer surgical procedure resulted directly in higher costs. The costs for preoperative work-up were not included. For further analysis, the total costs were subdivided into seven cost domains: general diagnostics (e.g., laboratory, pathology investigation, and microbiology), imaging (e.g., CT or MRI scans), outpatient clinic (e.g., visits to the emergency department or outpatient clinic within 90 days after discharge), clinical care (ward care), surgical (operating room and surgical supplies), ICU (critical care on ICU department), other costs (e.g., percutaneous drainage or similar procedures and bloodtransfusion). Analysis was carried out using 2017 costs in euros (EUR). Complicated postoperative course and complications were graded by the Clavien–Dindo classification [12,13]. The Comprehensive Complication Index was calculated with the online calculator from www.assessurgery.com (accessed on 3 October 2021) [14]. The primary outcome measure was in-hospital costs calculated in euros (EUR). POPF, delayed gastric emptying (DGE), bile leakage (BL), postoperative bleeding, and chylous leakage were defined as stated by the definitions of International Study Group for Pancreatic Surgery (ISGPS).

### 2.1. Statistics

Data were collected in Microsoft Office Excel 2010 and later transferred to SPSS Statistics 24 (IBM, Somer, NY, USA) for statistical analysis. Cost analysis was performed using R 3.6.1 (R Foundation for Statistical Computing, Vienna, Austria). Descriptive statistics were calculated as percentages, median, or mean, as appropriate. For the cost calculations, the interquartile range was used. A Mann–Whitney test was used for univariate analysis of continuous variables if data were not normally distributed. In case the data were normally distributed, an unpaired *t*-test was used. A Kruskal–Wallis test was used if data was not normally distributed to analyse continuous variables for multiple groups. In case of nominal/categorical variables, a Chi-squared test or a Fisher’s exact test was used; *p*-values < 0.05 were considered significant. Analyses were performed in SPSS v.24 (IBM, Somer, NY, USA)

### 2.2. Ethics

All data were retrieved from an existing database from the Dutch Pancreatic Cancer Audit (DPCA). Since we used the existing data register from the DPCA from the patients of our own hospital, no formal approval from the medical ethics committee was needed. Our local ethics committee waived the need for informed consent.

## 3. Results

### 3.1. Baseline Characteristics

A total of 160 patients (53% men, mean age 65 years, median age 67 years) who underwent a PD between January 2012 and November 2017 were included for analysis. Table 1 presents an overview of the baseline characteristics of the population. An uncomplicated course after PD was observed in 35 patients (21.9%). A total of 87 patients (54.4%) had minor complications, whereas 38 patients (23.8%) had severe (Clavien–Dindo IIIA or more) complications. UR occurred in 40 patients (25%), 18 (45%) of whom had minor, and 19 (47.5%) of whom had major complications during index admission.

A total of 125 (78.1%) of the patients underwent a PPPD, and 35 (21.9%) patients underwent a classic Whipple procedure. A vascular resection was performed in 26 patients (16.3%). In total, 97 patients (62.9%) underwent preoperative biliary stenting. In 80 patients (50%), the tumor originated from the pancreas. In 120 patients (70%), the pathology showed an adenocarcinoma. In total, 12 patients (7.5%) developed a clinically relevant (grade B/C) postoperative pancreatic fistula (CR-POPF) (see Table 1 and Table 2 for baseline characteristics).

### 3.2. Complicated Course

The median costs for patients with an uncomplicated course after PD were EUR 25.682 (IQR EUR 21.447–EUR 31.594). For patients with a complicated course (in general) after PD, the median costs were EUR 32.958 (IQR EUR 26.828–EUR 42.813, *p* = 0.001). For patients with minor complications, the median costs were EUR 30.316 (IQR EUR 24.717–EUR 36.291). For patients with major complications, these costs were EUR 42.664 (IQR EUR 35.942–EUR 55.407, *p* = 0.001). For patients with Clavien–Dindo grade IV complications, the median costs were EUR 90.0841 (IQR EUR 49.646–EUR 116.845). Complications after PD led to an increase in total hospital admission costs of EUR 4.634 (18%) in the case of minor complications; and in the case of major complications, EUR 16.982 (66.1%) (*p* = 0.001). (Table 3 and Table A1).

For patients with an uncomplicated postoperative course, the clinical care costs were EUR 5.845, while this amount increased to EUR 8.713 and EUR 16.276 for patients suffering minor and major complications, respectively (*p* = 0.001). Surgical costs seemed to be more or less the same for patients suffering no complications, minor complications, or major complications, and ranged from EUR 10.930 to EUR 12.210 (*p* = 0.642) (Table 3 and Table A1).

For patients with Clavien–Dindo grade IV complications, the median costs in every cost domain seemed to be higher compared to the group of patients with an uncomplicated course or less severe complications (*p* = 0.001–0.023). ICU costs, general diagnostic costs, imaging costs, and other costs seemed to relatively rise the most in case of Clavien–Dindo grade IV complications. (Table A2).

The mean and median days of hospital stay were 11.7 days and 11 days, respectively, for patients with an uncomplicated course after PD. For patients with a complicated course (in general) after PD, the mean and median days of hospital days were 23.3 days and 17 days, respectively (*p* = 0.001). The mean and median days of hospital stay for patients with minor complications were 17.1 days and 15 days, respectively. For patients with major complications, the mean and median days of hospital days were 37.6 days and 31 days, respectively (*p* = 0.001) (Table 3 and Table A1).

### 3.3. CCI Score

The median costs for patients with a CCI score of 0–30 ranged from EUR 27.350 to EUR 33.160. The median costs for patients with a CCI score between 30 and 50 ranged from EUR 40.246 to EUR 44.552. In cases of a CCI score of greater than 50, the median costs rose to EUR 93.311 (*p* = 0.001). The median costs for every different category rose when the CCI score increased. With clinical care costs, ICU costs and imaging costs increased the most in cases in which complications added up (Table 4 and Table A3). Overall, the median costs seemed to be higher in cases of an unplanned readmission, except for patients with a CCI score of >50, but the number of patients in this group was small (*n* = 6) (Figure A1). Figure 1A,B show the median total hospital costs by Clavien–Dindo classification and by the CCI score. Figure 1C shows a strong correlation (Pearson correlation coefficient = 0.77) between the CCI score and the log of the total costs. Therefore, hospital costs were directly related to the CCI score, with every single complication adding up in total hospital costs.

The mean and median days of hospital stay for patients with a CCI score of 0–30 ranged from 12.4 to 19 days and 12 to 18 days, respectively. The mean and median days of hospital stay for patients with a CCI score between 30 and 50 ranged from 28.4 to 34.6 days and 26 to 31 days, respectively. In cases of a CCI score of greater than 50, the mean and median days of hospital stay were 58.7 and 53 days, respectively. Hospital stays seemed to be longer when the CCI score increased (*p* = 0.001) (Table 4 and Table A3).

### 3.4. Unplanned Readmissions

The median total hospital costs of patients with one or more UR were EUR 41.199 (IQR EUR 33.030–EUR 45.834, *p* = 0.001). For patients with UR and minor complications, the median costs were EUR 40.875 (IQR EUR 30.473–EUR 43.260). For patients with UR and major complications, these costs were EUR 41.739 (IQR EUR 36.211–EUR 52.347, *p* = 0.076). A UR was associated with an increase of EUR 12.567 (44%) in total hospital costs compared to the group of patients without readmissions (*p* = 0.001). This increase was mainly explained by an increase in general diagnostics costs, imaging costs, clinical care costs, and other costs, as these costs were higher in cases of a UR (*p* = 0.001–0.002) (Table 5 and Table A4, Figure A2). The median hospital costs in patients without UR versus patients with UR categorized by complication type (no complications, minor complications and major complications) can be found in Table 6 and Table A5.

The mean and median days of hospital stay were 17.1 days and 13 days, respectively, for patients with no readmissions. For patients with readmissions, the mean and median days of hospital days were 31.7 days and 26 days, respectively (*p* = 0.001) (Table 5 and Table A4).

## 4. Discussion

This single academic center study revealed that complications after PD led to a EUR 4.634–EUR 16.982 (18–66%) increase in hospital admission costs. These costs were mainly due to an increase in clinical care costs. In cases of Clavien–Dindo grade IV complications, the median costs in every cost domain were higher compared to an uncomplicated course or less-severe complications (Clavien–Dindo I–III). In particular, ICU costs, general diagnostic costs, imaging costs, and other costs seemed to relatively rise the most compared to less-severe complications (Clavien–Dindo I–III). A UR was associated with a EUR 12.567 (44%) increase in hospital costs. Log hospital costs seemed to be directly related to the CCI score, with a Pearson correlation coefficient of 0.77. Clinical care costs, ICU costs, and imaging costs especially seemed to rise the most in case complications add up.

With healthcare costs increasing annually worldwide, hospital cost containment will be one of the biggest challenges. One of the reasons for the increase in healthcare costs is the ageing population, resulting in patients with multimorbidity, and more often a complicated course after surgery. Postoperative complications have a high impact on quality of life and healthcare costs for society. A PD is a complex procedure often performed in specialized pancreatic centres. Although the mortality rate of pancreatic resections has been reduced over the past decades to less than 5% by means of centralization, leading to higher volumes and lower failure to rescue rates, the overall morbidity remains around 30% to 60% [6,7,8] Well-known complications after pancreatic resection include PF, DGE, haemorrhage, and deep and superficial surgical site infections [6,7,8]. The number of general nonsurgical complications such as pulmonary and cardiovascular complications rose in general with increasing age. A complicated postoperative course may lead to readmissions, and both were associated with high hospital costs. In the literature, there were only a few studies that analysed hospital costs after PD. Therefore, we aimed to gain an understanding in the costs of complications and unplanned readmissions in patients who underwent a PD. Previous studies have reported that a complicated course after PD increased hospital costs substantially, and may even lead to double or triple in-hospital costs, depending on the severity of complications [1,3,9]. Our study showed comparable results, but the effect was less pronounced. In our study, the median hospital costs for patients without complications were EUR 25.682. For patients with severe complications, the median costs rose to EUR 42.664. An UR was associated with a EUR 12.567 increase in hospital costs. In a study by Santema et al., the median hospital costs for patients without complications were EUR 17.482. For patients with grade I complications (defined as CD I–IIIa) the median hospital costs were EUR 28.380, and for grade II complications (defined as CD > IIIb), the median hospital costs were EUR 57.060 [1]. An American study by Enestvedt et al. showed comparable results to the study of Santema at al., with median hospital costs for patients with severe complications being significantly higher than for those without (USD 56.224 vs. USD 29.038) [3]. A study by Staiger at al. showed a strong correlation (Pearson correlation coefficient = 0.70) between log hospital costs and the CCI score [10]. Our study showed even a better correlation, with a Pearson correlation coefficient of 0.77. As complications are associated with severely increased hospital costs, continuous efforts are needed to reduce complications aiming to lower hospital costs for PD. Since our study and previous studies have shown that reinterventions under full anaesthesia or ICU admission (CD IIIb–IV) were associated with up to a threefold increase in hospital costs, early detection and management of complications is important to reduce both the length of hospital stays and hospital costs [1,3].

The following three strategies could reduce costs. Early detection and treatment of complications might prevent worsening of severity of complications. In 2018, the DPCG started the PORSCH trial, a nationwide stepped-wedge cluster randomized trial [15]. The aim of this trial was to evaluate if implementation of a best-practice algorithm for postoperative care focusing on early detection and step-up management of POPF could result in a lower rate of major complications and mortality after pancreatic resection, as compared to the current practice. Since hospital costs seemed to increase substantially after a complicated course after surgery, and every single complication added up in costs, as shown by the relationship between hospital costs and CCI, efforts to reduce complications and to minimize risk factors for complications will most likely be cost-effective soon. Additionally, insight into hospital costs could be a surrogate marker for quality control of surgical outcome.

Another strategy to reduce complication and thereby lower healthcare costs might be the introduction of surgical care bundles (SCBs) and enhanced recovery programs (ERPs) in pancreatic surgery. Over the past decade, the introduction of SCBs in colorectal and liver surgery have proven to significantly reduce surgical site infections (SSI) [16,17]. Various components of an SCB usually include hair removal prior to surgery, perioperative antibiotic prophylaxis, and perioperative normoglycemia and normothermia [17,18]. For pancreatic surgery, an overall SSI rate of around 20–30% was described in the literature [19,20]. In a study by Lawrence at al., an SCB for pancreatic surgery was implemented, and was found to decrease the SSI rate from 22% to 11% [20]. Furthermore, ERP implementation seems to be associated with a 50% reduction in risk for postoperative complications, as well as a reduction in the length of hospital stay by up to 2.4 days [18,21,22,23]. Between 2013 and 2017, ERPs and SCBs were not yet implemented in our hospital. The combination of implementing SCBs and ERPs in pancreatic surgery might lead to a reduction in complications and thereby healthcare costs, but future studies are needed to confirm this hypothesis.

A third strategy to reduce complication and prevent worsening of severity of complications and thereby lower healthcare costs might be prehabilitation. Prehabilitation is a way to preoperatively optimize modifiable risk factors such as poor physical fitness, malnutrition, low mental resilience, iron deficiency anaemia, and smoking and/or drinking alcohol [23,24,25]. This can be achieved by physical exercise/training with a physiotherapist, optimizing a patient’s diet with a dietician, and psychological support and/or coaching prior to surgery. In addition, optimizing hemoglobin levels prior to surgery by iron infusion is effective in reducing length of hospital stay and number of red cell transfusions [24,25]. Prehabilitation has shown in several studies to contribute to reduction in postoperative complications, faster physical recovery after surgery, and a shorter hospital stay, and is thought to lower healthcare costs in the end [24,25,26]. Previous studies have shown that prehabilitation reduces the overall and pulmonary morbidity after surgery [27]. Future studies to prove the cost-effectiveness of prehabilitation are needed, but since our study, as well as previous studies, showed that hospital were directly related to any increase in CCI, it is plausible that efforts to prehabilitate patients before PD or surgery in general might soon be cost-effective.

Our study had potential limitations. Only healthcare costs incurred in our institute were available for analysis. Since our hospital is a tertiary referral center for pancreatic surgery, some patients might have been readmitted in a local hospital nearby instead of the index hospital. This means that some costs might not be included in our study, meaning the overall costs of a complications and UR after PD might be even higher than previously stated. Nevertheless, we expect that the number of patients who did not go to our hospital for their complication was low, since patients were advised to go to the index hospital in case problems occurred. Another limitation was that we only examined hospital costs. Costs from an eventual admission in a nursing home or rehabilitation center after hospital discharge were not included. This means that the overall costs of a complications and UR after PD might be even higher than previously stated.

The strength of our study was that we used a larger study population (160 patients) and a more detailed hospital costs analysis compared to previous studies. Furthermore, our study and detailed cost overview served as a substantiation for a business model for a multimodal prehabilitation program in our hospital [11].

## 5. Conclusions

Our study showed that complications after PD led to a EUR 4.634–EUR 16.982 (18–66%) increase in hospital admission costs, mainly due to an increase in costs of clinical care. A UR was associated with a EUR 12.567 (44%) increase in hospital costs. Hospital costs seemed to be directly related to the CCI score. Therefore, every reduction in postoperative complications will contribute to an increased cost-effectiveness.

## Figures and Tables

**Figure 1 cancers-13-06271-f001:**
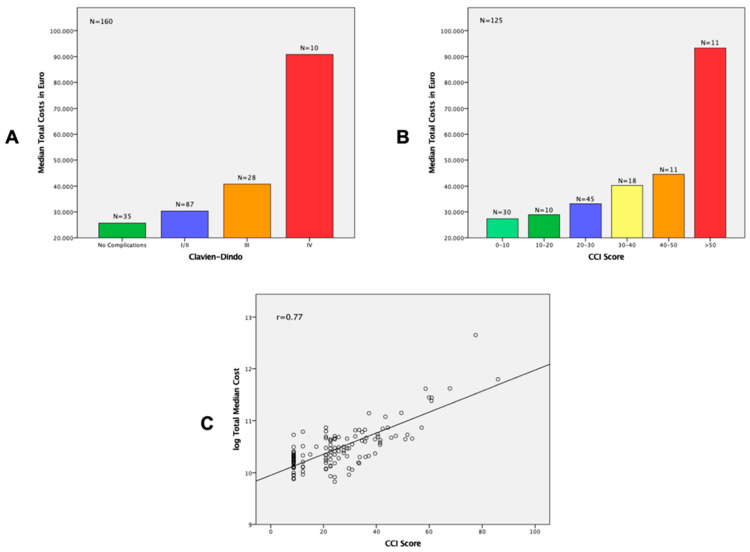
Cost analysis for Clavien-Dindo classification and CCI score. (**A**) Median total hospital costs (uncomplicated course and complicated course by Clavien–Dindo classification) (*p* = 0.001) (**B**) Median total hospital costs (complicated course by CCI score) (*p* = 0.001) (**C**) Correlation of CCI and log of total costs with Pearson correlation coefficient (r_pears_).

**Table 1 cancers-13-06271-t001:** Baseline characteristics.

	Study Cohort (*n* = 160)
Age (mean, median, IQR)	65; 67; 59–72
Sex (male), no. (%).	85 (53.1%)
BMI (mean, median, IQR)	26.0; 25.5; 22.8–28.4
<20	8 (5%)
20–30	128 (80%)
>30	24 (15%)
ECOG score	
0	101 (63.1%)
1	50 (31.3%)
2	8 (5%)
3	1 (0.6%)
ASA score	
1	16 (10%)
2	113 (70.6%)
3	31 (19.4%)
Comorbidities	
Cardiac	35 (21.9%)
Vascular	49 (30.6%)
Diabetes	28 (17.5%)
Pulmonary	15 (9.4%)
No complications	35 (21.9%)
Complications	125 (78.1%)
Clavien–Dindo I/II	87 (54.4%)
Clavien–Dindo IIIa	23 (14.4%)
Clavien–Dindo IIIb	5 (3.1%)
Clavien–Dindo IV	10 (6.3%)
Unplanned readmissions	40 (25%)
No complications	3 (7.5%)
Clavien–Dindo I/II	18 (45%)
Clavien–Dindo III	16 (40%)
Clavien–Dindo IV	3 (7.5%)
CCI score	
0–10	30 (18.8%)
10–20	10 (6.3%)
20–30	45 (28.1%)
30–40	18 (11.3%)
40–50	11 (6.9%)
>50	11 (6.9%)
Mortality, no. (%)	0 (-)
Pathology	
Adenocarcinoma	112 (70%)
Neuroendocrine neoplasm	13 (8.1%)
IPMN (Intraductal Papillary Mucinous Neoplasm)	9 (5.6%)
Mucinous Cystic Neoplasm	1 (0.6%)
Adenoma Intestinal Type	7 (4.4%)
Serous Cystadenoma	1 (0.6%)
Chronic Pancreatitis	4 (2.5%)
Other/Unknown	13 (8.1%)
Origin of tumor	
Pancreas	80 (50%)
Distal bile duct	24 (15%)
Ampulla of Vater	22 (13.8%)
Duodenum	16 (10%)
Other/Unknown	18 (11.3%)
Type of resection	
Classic Whipple	35 (21.9%)
PPPD	125 (78.1%)
Vascular resection	26 (16.3%)
Preoperative biliary stenting	
No	62 (38.8%)
ERCP	91 (56.9%)
PTC	6 (3.8%
Unknown	1 (0.6%)

**Table 2 cancers-13-06271-t002:** Baseline characteristics: complications vs. no complications and readmissions vs. no readmissions.

	Study Cohort All Patients (*n* = 160)	No Complications (*n* = 35)	Complicated Course (*n* = 125)	No Readmissions (*n* = 120)	Unplanned Readmissions (*n* = 40)
Age (mean, median, IQR)	65; 67; 59–72	64; 67; 56–73	65; 67; 60–72	65; 67; 59–73	65; 67; 60–71
Sex (male), no. (%).	85 (53.1%)	23 (65.7%)	62 (49.6%)	67 (55.8%)	18 (45%)
BMI (mean, median, IQR)	26.0; 25.5; 22.8–28.4	25.1; 24.6; 22.9–26.6	26.3; 25.7; 22.6–28.7	26.2; 25.3; 22.9–28.7	25.7; 26.0; 22.1–27.3
<20	8 (5%)				
20–30	128 (80%)				
>30	24 (15%)				
ECOG score					
0	101 (63.1%)	25 (71.4%)	76 (60.8%)	81 (67.5%)	20 (50%)
1	50 (31.3%)	9 (25.7%)	41 (32.8%)	33 (27.5%)	17 (42.5%)
2	8 (5%)	1 (2.9%)	7 (5.6%)	5 (4.2%)	3 (7.5%)
3	1 (0.6%)	0 (-)	1 (0.8%)	1 (0.8%)	0 (-)
ASA score					
1	16 (10%)	4 (11.4%)	12 (9.6%)	15 (12.5%)	1 (2.5%)
2	113 (70.6%)	26 (74.3%)	87 (69.6%)	88 (73.3%)	25 (62.5%)
3	31 (19.4%)	5 (14.3%)	26 (20.8%)	17 (14.2%)	14 (35%)
Comorbidities					
Cardiac	35 (21.9%)	4 (11.4%)	31 (24.8%)	24 (20%)	11 (27.5%)
Vascular	49 (30.6%)	8 (22.9%)	41 (32.8%)	41 (34.2%)	8 (20%)
Diabetes	28 (17.5%)	11 (31.4%)	17 (13.6%)	25 (20.8%)	3 (7.5%)
Pulmonary	15 (9.4%)	4 (11.4%)	11 (8.85)	13 (10.8%)	2 (55)
Pathology					
Adenocarcinoma	112 (70%)	26 (74.3%)	86 (68.8%)	86 (71.7%)	26 (65%)
Neuroendocrine neoplasm	13 (8.1%)	0 (-)	13 (10.4%)	9 (7.5%)	4 (10%)
IPMN	9 (5.6%)	4 (11.4%)	5 (4%)	7 (5.8%)	2 (5%)
Mucinous cystic neoplasm	1 (0.6%)	0 (-)	1 (0.8%)	1 (0.8%)	0 (-)
Adenoma intestinal type	7 (4.4%)	1 (2.9%)	6 (4.8%)	4 (3.3%)	3 (7.5%)
Serous cystadenoma	1 (0.6%)	0 (-)	1 (0.8%)	0 (-)	1 (2.5%)
Chronic pancreatitis	4 (2.5%)	2 (5.7%)	2 (1.6%)	4 (3.3%)	0 (-)
Other/Unknown	13 (8.1%)	2 (5.8%)	11 (8.8%)	9 (7.5%)	4 (10%)
Origin of tumor					
Pancreas	80 (50%)	19 (54.3%)	61 (48.8%)	62 (51.7%)	18 (45%)
Distal bile duct	24 (15%)	6 (17.1%)	18 (14.4%)	19 (15.8%)	5 (12.5%)
Ampulla of Vater	22 (13.8%)	4 (11.4%)	18 (14.4%)	17 (14.2%)	5 (12.5%)
Duodenum	16 (10%)	4 (11.4%)	12 (9.6%)	13 (10.8%)	3 (7.5%)
Other/Unknown	18 (11.3%)	2 (5.8%)	16 (12.8%)	9 (7.6%)	9 (22.5%)
Type of resection					
Classic Whipple	35 (21.9%)	5 (14.3%)	30 (24%)	22 (18.3%)	13 (32.5%)
PPPD	125 (78.1%)	30 (85.7%)	95 (76%)	98 (81.7%)	27 (67.5%)
Vascular resection	26 (16.3%)	7 (20%)	19 (15.2%)	20 (16.7%)	6 (15%)
Preoperative biliary stenting					
No	62 (38.8%)	11 (31.4%)	51 (40.8%)	44 (36.7%)	18 (45%)
ERCP	91 (56.9%)	23 (65.7%)	68 (54.4%)	70 (58.3%)	21 (52.5%)
PTC	6 (3.8%	1 (2.9%)	5 (4%)	5 (4.2%)	1 (2.5%)
Unknown	1 (0.6%)	0 (-)	1 (0.8%)	1 (0.8%)	0 (-)
No complications	35 (21.9%)	n.a.	0 (-)	32 (26.7%)	3 (7.5%)
Complications	125 (78.1%)		125	88 (73.3%)	37 (92.5%)
Clavien–Dindo I/II	87 (54.4%)		87 (69.6%)	69 (57.5%)	18 (45%)
Clavien–Dindo IIIa	23 (14.4%)		23 (18.4%)	9 (7.5%)	14 (35%)
Clavien–Dindo IIIb	5 (3.1%)		5 (4%)	3 (2.5%)	2 (5%)
Clavien–Dindo IV	10 (6.3%)		10 (8%)	7 (5.8%)	3 (7.5%)
CCI score		n.a.			
0–10	30 (18.8%)		30 (24%)	29 (24.2%)	1 (2.5%)
10–20	10 (6.3%)		10 (8%)	10 (8.3%)	0 (-)
20–30	45 (28.1%)		45 (36%)	31 (25.8%)	14 (35%)
30–40	18 (11.3%)		18 (14.4%)	7 (5.8%)	11 (27.5%)
40–50	11 (6.9%)		11 (8.8%)	6 (5%)	5 (12.5%)
>50	11 (6.9%)		11 (8.8%)	5 (4.2%)	6 (15%)
Postoperative Pancreatic Fistula (POPF)		n.a.			
No	144 (90%)		109 (87.2%)	110 (91.7%)	34 (85%)
Biochemical leakage	4 (2.5%)		4 (3.2%)	4 (3.3%)	0 (-)
Grade B	9 (5.6%)		9 (7.2%)	5 (4.2%)	4 (10%)
Grade C	3 (1.9%)		3 (2.4%)	1 (0.8%)	2 (5%)
Delayed Gastric Emptying		n.a.			
No	120 (75%)		85 (68%)	95 (79.2%)	25 (62.5%)
Grade A	11 (6.9%)		11 (8.8%)	8 (6.7%)	3 (7.5%)
Grade B	14 (8.8%)		14 (11.2%)	9 (7.5%)	5 (12.5%)
Grade C	15 (9.4%)		15 (12%)	8 (6.7%)	7 (17.5%)
Bile leakage		n.a.			
No	159 (99.4%)		124 (99.2%)	119 (99.2%)	40 (100%)
Grade A	0 (-)		0 (-)	0 (-)	0 (-)
Grade B	1 (0.6%)		1 (0.8%)	1 (0.8%)	0 (-)
Grade C	0 (-)		0 (-)	0 (-)	0 (-)
Bleeding		n.a.			
No	151 (94.4%)		116 (92.8%)	115 (95.8%)	36 (90%)
Grade A	1 (0.6%)		1 (0.8%)	0 (-)	1 (2.5%)
Grade B	6 (3.8%)		6 (4.8%)	5 (4.2)	1 (2.5%)
Grade C	2 (1.3%)		2 (1.6%)	0 (-)	2 (5%)
Chylous leakage		n.a.			
No	113 (70.6%)		78 (62.4%)	84 (70%)	29 (72.5%)
Grade A	36 (22.5%)		36 (28.8%)	29 (24.2%)	7 (17.5%)
Grade B	11 (6.95)		11 (8.8%)	7 (5.8%)	4 (10%)
Grade C	0 (-)		0 (-)	0 (-)	0 (-)
Pneumonia		n.a.			
No	158 (98.8%)		123 (98.4%)	119 (99.2%)	39 (97.5%)
Yes	2 (1.3%)		2 (1.6%)	1 (0.8%)	1 (2.5%)
Wound infection		n.a.			
No	139 (86.9%)		104 (83.2%)	107 (89.2%)	32 (80%)
Yes	21 (13.1%)		21 (16.8%)	13 (10.8%)	8 (20%)
Hospital stay in days (mean, median, IQR)	20.8; 15.5; 11.0–24.8	11.7; 11; 9–13	23.3; 17; 13–27	17.1; 13; 11.0–17.8	31.7; 26.5; 20.0–40.5

**Table 3 cancers-13-06271-t003:** Median hospital costs of uncomplicated vs. complicated course divided by minor and major complications.

Median Costs (IQR)	No Complications *n* = 35	Complicated Course (in General) *n* = 125	Minor Complications (CDI/II) *n* = 87	Major Complications (CD > III) *n* = 38
Total costs	EUR 25.682 (EUR 21.447–EUR 31.594)	EUR 32.958 (EUR 26.828–EUR 42.813)	EUR 30.316 (EUR 24.717–EUR 36.291)	EUR 42.664 (EUR 35.942–EUR 55.407)
General diagnostics	EUR 1.571 (EUR 1.308–EUR 1.973)	EUR 2.113 (EUR 1.669–EUR 3.126)	EUR 1.874 (EUR 1.529–EUR 2.652)	EUR 2.740 (EUR 2.112–EUR 3.813)
Imaging	EUR 54 (EUR 54–EUR 582)	EUR 346 (EUR 108–EUR 1.157)	EUR 162 (EUR 108–EUR 635)	EUR 891 (EUR 520–EUR 2.268)
Outpatient clinic	EUR 0 (EUR 0–EUR 85)	EUR 105 (EUR 0–EUR 625)	EUR 0 (EUR 0–EUR 468)	EUR 486 (EUR 56–EUR 964)
Clinical care	EUR 5.845 (EUR 5.206–EUR 7.014)	EUR 9.898 (EUR 6.958–EUR 15.623)	EUR 8.713 (EUR 6.402–EUR 11.630)	EUR 16.276 (EUR 12.186–EUR 26.363)
ICU	EUR 4.407 (EUR 4.312–EUR 4.407)	EUR 4.407 (EUR 4.312–EUR 6.610)	EUR 4.407 (EUR 4.312–EUR 6.468)	EUR 6.468 (EUR 4.407–EUR 13.475)
Surgical	EUR 10.930 (EUR 8.143–EUR 13.845)	EUR 11.875 (EUR 8.586–EUR 14.648)	EUR 11.585 (EUR 8.679–EUR 14.634)	EUR 12.210 (EUR 8.420–EUR 14.983)
Other costs	EUR 498 (EUR 183–EUR 2.256)	EUR 2.207 (EUR 743–EUR 2.926)	EUR 2.134 (EUR 586–EUR 2.670)	EUR 2.906 (EUR 1.159–EUR 5.003)
Hospital stay in days (mean, median, IQR)	11.7; 11; 9–13	23.3; 17; 13–27	17.1; 15; 12–20	37.6; 31; 20.8–45.3

**Table 4 cancers-13-06271-t004:** Median hospital costs (complicated course by CCI score).

Median Costs (IQR)	CCI 0–10 *n* = 30	CCI 10–20 *n* = 10	CCI 20–30 *n* = 45	CCI 30–40 *n* = 18	CCI 40–50 *n* = 11	CCI > 50 *n* = 11
Total costs	EUR 27.350 (EUR 24.411–EUR 30.388)	EUR 28.921 (EUR 24.046–EUR 36.361)	EUR 33.160 (EUR 27.311–EUR 40.875)	EUR 40.246 (EUR 29.799–EUR 48.404)	EUR 44.552 (EUR 41.018–EUR 52.347)	EUR 93.311 (EUR 45.563–EUR 111.537)
General diagnostics	EUR 1.658 (EUR 1.231–EUR 2.323)	EUR 1.764 (EUR 1.302–EUR 1.930)	EUR 1.986 (EUR 1.704–EUR 2.934)	EUR 2.701 (EUR 1.857–EUR 3.549)	EUR 2.714 (EUR 2.405–EUR 3.518)	EUR 6.406 (EUR 2.880–EUR 8.996)
Imaging	EUR 108 (EUR 54–EUR 220)	EUR 166 (EUR 108–EUR 2.575)	EUR 292 (EUR 162–EUR 558)	EUR 702 (EUR 259–EUR 1.062)	EUR 1.186 (EUR 612–EUR 2.126)	EUR 2.954 (EUR 975–EUR 4.568)
Outpatient clinic	EUR 0 (EUR 0–EUR 0)	EUR 85 (EUR 0–EUR 196)	EUR 182 (EUR 0–EUR 630)	EUR 634 (EUR 0–1.075)	EUR 468 (EUR 167–EUR 936)	EUR 650 (EUR 468–EUR 1.364)
Clinical care	EUR 6.429 (EUR 6.222–EUR 7.744)	EUR 8.721 (EUR 6.236–EUR 9.683)	EUR 10.580 (EUR 7.589–EUR 13.929)	EUR 14.550 (EUR 11.044–EUR 23.785)	EUR 18.125 (EUR 10.425–EUR 25.690)	EUR 28.382 (EUR 15.700–EUR 49.112)
ICU	EUR 4.407 (EUR 4.312–EUR 4.407)	EUR 5.508 (EUR 4.312–EUR 7.114)	EUR 4.407 (EUR 4.312–EUR 6.539)	EUR 4.407 (EUR 4.312–EUR 6.504)	EUR 6.468 (EUR 4.407–EUR 6.610)	EUR 15.423 (EUR 8.813–EUR 48.474)
Surgical	EUR 11.222 (EUR 8.545–EUR 14.404)	EUR 10.807 (EUR 9.185–EUR 13.694)	EUR 12.189 (EUR 8.384–EUR 14.883)	EUR 9.235 (EUR 6.384–EUR 12.327)	EUR 12.544 (EUR 8.492–EUR 14.554)	EUR 16.235 (EUR 11.067–EUR 18.614)
Other costs	EUR 2.159 (EUR 369–EUR 2.558)	EUR 2.116 (EUR 447–EUR 2.889)	EUR 1.514–EUR 739–EUR 2.607)	EUR 2.119 (EUR 909–EUR 3.817)	EUR 3.200 (EUR 2.900–EUR 4.996)	EUR 6.035 (EUR 1.971–EUR 11.878)
Hospital stay in days (mean, median, IQR)	12.4; 12; 10–14	14.9; 16; 11.5–17.0	19; 18; 13.5–23.5	28.4; 26; 18.0–39.8	34.6; 31; 25–44	58.7; 53; 31–72

**Table 5 cancers-13-06271-t005:** Median hospital costs in patients without readmissions vs. patients with unplanned readmissions.

Median Costs (IQR)	No Readmissions *n* = 120	Unplanned Readmissions *n* = 40
Total costs	EUR 28.552 (EUR 24.321–EUR 35.324)	EUR 41.199 (EUR 33.030–EUR 45.834)
General diagnostics	EUR 1.885 (EUR 1.471–EUR 2.643)	EUR 2.494 (EUR 1.757–EUR 3.520)
Imaging	EUR 158 (EUR 54–EUR 620)	EUR 684 (EUR 375–EUR 1.291)
Outpatient clinic	EUR 0 (EUR 0–EUR 137)	EUR 714 (EUR 468–EUR 1.228)
Clinical care	EUR 7.534 (EUR 5.845–EUR 10.289)	EUR 15.943 (EUR 11.667–EUR 23.644)
ICU	EUR 4.407 (EUR 4.312–EUR 6.610)	EUR 4.407 (EUR 4.312–EUR 6.610)
Surgical	EUR 11.530 (EUR 8.274–EUR 14.655)	EUR 11.875 (EUR 8.336–EUR 14.541)
Other costs	EUR 1.690 (EUR 393–EUR 2.602)	EUR 2.763 (EUR 1.113–EUR 3.555)
Hospital stay in days (mean, median, IQR)	17.1; 13; 11.0–17.8	31.7; 26.5; 20.0–40.5

**Table 6 cancers-13-06271-t006:** Median hospital costs in patients without readmissions versus patients with unplanned readmissions (minor complications vs. major complications).

	No Readmissions *n* = 120			Unplanned Readmissions *n* = 40		
Median Costs (IQR)	No Complications *n* = 32	Minor Complications (CDI/II) *n* = 69	Major Complications (CD > III) *n* = 19	No Complications *n* = 3	Minor Complications (CDI/II) *n* = 18	Major Complications (CD > III) *n* = 19
Total costs	EUR 25.416 (EUR 21.366–EUR 29.721)	EUR 28.336 (EUR 24.366–EUR 33.282)	EUR 44.033 (EUR 32.206–EUR 64.586)	EUR 32.800 (n.a.)	EUR 40.875 (EUR 30.473–EUR 43.260)	EUR 41.739 (EUR 36.211–EUR 52.347)
General diagnostics	EUR 1.530 (EUR 1.279–EUR 1.962)	EUR 1.862 (EUR 1.482–EUR 2.611)	EUR 2.714 (EUR 2.162–EUR 4.607)	EUR 1.754 (n.a.)	EUR 2.388 (EUR 1.728–EUR 3.286)	EUR 2.880 (EUR 1.865–EUR 3.664)
Imaging	EUR 54 (EUR 54–EUR 261)	EUR 108 (EUR 80–EUR 408)	EUR 935 (EUR 498–EUR 2.543)	EUR 2.548 (n.a.)	EUR 502 (EUR 284–EUR 857)	EUR 858 (EUR 580–EUR 1.293)
Outpatient clinic	EUR 0 (EUR 0–EUR 85)	EUR 0 (EUR 0–118)	EUR 158 (EUR 0–EUR 565)	EUR 468 (n.a.)	EUR 689 (EUR 520–EUR 1.077)	EUR 718 (EUR 468–EUR 1.280)
Clinical care	EUR 5.831 (EUR 5.206–EUR 6.960)	EUR 7.579 (EUR 6.375–EUR 9.387)	EUR 13.946 (EUR 10.420–EUR 21.041)	EUR 9.936 (n.a.)	EUR 12.861 (EUR 11.148–EUR 19.062)	EUR 22.102 (EUR 15.652–EUR 30.284)
ICU	EUR 4.407 (EUR 4.312–EUR 4.407)	EUR 4.407 (EUR 4.312–EUR 6.468)	EUR 6.610 (EUR 6.468–EUR 15.423)	EUR 4.312 (n.a.)	EUR 4.359 (EUR 4.312–EUR 4.958)	EUR 4.407 (EUR 4.407–EUR 8.813)
Surgical	EUR 11.202 (EUR 7.971–EUR 13.841)	EUR 10.821 (EUR 8.143–EUR 14.567)	EUR 13.683 (EUR 11.073–EUR 16.305)	EUR 10.842 (n.a.)	EUR 12.221 (EUR 9.745–EUR 14.802)	EUR 9.613 (EUR 6.384–EUR 13.845)
Other costs	EUR 465 (EUR 175–EUR 2.246)	EUR 1.827 (EUR 470–EUR 2.501)	EUR 2.785 (EUR 1.354–EUR 5.024)	EUR 2.073 (n.a.)	EUR 2.757 (EUR 875–EUR 3.409)	EUR 2.988 (EUR 1.110–EUR 3.660)
Hospital stay in days (mean, median, IQR)	10.9; 11; 9–12	14.9; 14; 12–17	35.5; 27; 19–39	20; 19; n.a.	25.2; 22.5; 18.8–32.5	39.6; 35; 25–49

## Data Availability

An anonymized version of our database can be provided upon request.

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
