# Peer review of "The Costs of Complications and Unplanned Readmissions after Pancreatoduodenectomy for Pancreatic and Periampullary Tumors: Results from a Single Academic Center"

_cancers, 2021, doi:10.3390/cancers13246271_

Round 1

Reviewer 1 Report

This is an single-center original study regarding the costs of complications and unplanned readmissions after pancreatoduodenectomy for pancreatic and periampullary tumors.

In abstract, the study period should be presented. In results, p values should be added. Median costs for an uncomplicated course were €25 682 and for a complicated course €32 958 (p=?). The median costs for minor complications were €30 316 and for severe complications €42 664 (p=?).

In introduction: in the sentence: A pancreatoduodenectomy (PD) is a complex procedure usually performed in case 37 of malignant tumours of the pancreatic head or surrounding structures: "surrounding structures" should be precised.

In material an methods: in the sentence: All consecutive adult patients who underwent PD (classic Whipple 66 procedure/pylorus resecting pancreatoduodenectomy (PRPD) or pylorus preserving pancreatoduodenectomy (PPPD)) between 2013 and 2017 at our institute were included in the study, the number of patients should be presented.

In Table 1, there are 3 unplanned readmissions with no complications. What were the reasons of these radmissions?

In Table 1, regarding pathology, "benign" and "other" should be precised.

In Table 1, regarding origin of tumor, and "other" should be precised. Why the tumor origin was unknown in 10 patients?

In Tables 2-4, types of statistical tests and p values should be presented.

Regarding Figure 1. A. Median total hospital costs (Uncomplicated course and complicated course by Clavien-Dindo classification); B. Median total hospital costs (Complicated course by CCI score), statistical difference between these groups should be analyzed. The post-hoc tests for finding statistical difference should be used.

The general characteristics (including age, gender, BMI, comorbidities, tumor location and histopathology including TNM classification, preoperative biliary stenting, type of PD, vacular resection (if it was performed), ASA, duration of hospitalization) should be compared between complication vs. non-complications and readmission vs. non-readmission groups.

In results, all types complications should be presented (such as pancreatic fistula type A, B, C, wound infection, bile leak, DGE, etc. ). Also, correlation between types of complications and costs would be interesting fo readers.

What about length of hospital stay? Was a difference between patients with and without complications or according to Clavien-Dindo classification or complication types? The association betwen duration of hospitalization and complications types and classifications as well as duration of hospitalization and costs should be analyzed.

What was the impact of comorbidities on complications and costs? It should be analyzed.

Additional, advanced statistical methods (regression analyses) should be used in this study. Cost ratio for the increase in costs related to complications should be calculated using regression analysis. Multivariate linear regression was used to deter-mine The association between specific complications and costs should be calculated using multivariate linear regression analysis.
Limitations of the study are presented. What about strengths? The authors have not explained what was the novelty of their study. The presented analyses are interesting and precise, but it is generally known that complications are associated with higher hospitalization costs because of longer hospital stay and additional conservative or surgical treatment.

Author Response

We would like to thank the reviewers for their good comments and suggestions. We think it improved the overall manuscript significantly. We tried to process as much of the comments and suggestions as possible. In the abstract we have mentioned the study period and p-values were added. In the introduction the ‘surrounding structures’ were specified. In the material and methods the number of patients were presented. The 3 unplanned readmissions with no complications were patients that were admitted for observation due to malaise, pain or nausea complaints and were discharged rapidly after complaints disappeared. In table 1 regarding pathology ‘benign’ was further specified. In table 1 regarding origin of tumor. Other/unknown means that data was missing in our database or it was scored as other in the database. In table 2-4 the types of statistical test and p-values were added (see appendix section). In the description of figure 1A and 1B p-values were added (figure 1A and 1B are based on the results of table 2b and 3). Preoperative biliary stenting, type of PD, vascular resection and duration of hospitalization were added to the baseline characteristics table for complication vs. non-complications and readmission vs. no-readmission as well as all types of complications. Length of hospital stay was also added to table 1b, 2a,2b, 3 and 4. The strengths of the study were described in the discussion section. The strength of our study is that we have a larger study population (160 patients) and a more detailed hospital costs analysis compared to previous studies. In consultation with our statistician we decided not to perform a regression analysis but to look at the costs by Clavien-Dindo classification and CCI score. We think we have provided a detailed and precise cost overview for complications by Clavien-Dindo classification as well as the CCI score and readmissions. We think we that we have processed the reviewers comments and suggestions adequately and we hope it is to the satisfaction of the reviewers.

Reviewer 2 Report

Linnemann et al. aimed to create understanding in the costs of complications and unplanned readmission (UR)  in patients who underwent a pancreaticoduodenectomy (PD). In 160 patients operated in their Institute, the Authors showed that postoperative complications and unplanned readmissions significantly increased hospital costs and concluded that reduction of complications will lead to cost-effectiveness.

The problem of hospital cost in such complex surgery has been previously investigated and the results of this study confirm previous reports. Pancreatic surgeons are aware that complications after pancreatic resection may increase lenght of hospitalization and costs and every efforts should be directed to prevent or early recognize the complication. Since the most frequent and dangerous complication after PD is pancreatic fistula, it is not clear which is the strategy to reduce its cost. I think that this is the major problem.

Author Response

(The authors gave the same response as above.)

Round 2

Reviewer 1 Report

It seems that authors improved their manuscript according to the most my commnets and suggestions. I would like to ask the authors to explain, point by point, the details of the revisions to the manuscript and your responses to mycomments (comment above : answer below)
If they found it impossible to address certain comments in the review reports, an explanation in their rebuttal should be included.

Author Response

Rebuttal:

We would like to thank the reviewers for their good comments and suggestions. We think it improved the overall manuscript significantly. We tried to process as much of the comments and suggestions as possible. Hereby a point by point rebuttal. In the attachement the latest version of the manuscript (with all the changes marked in yellow) can be found.

This is an single-center original study regarding the costs of complications and unplanned readmissions after pancreatoduodenectomy for pancreatic and periampullary tumors.

  1. In abstract, the study period should be presented. In results, p values should be added. Median costs for an uncomplicated course were €25 682 and for a complicated course €32 958. The median costs for minor complications were €30 316 and for severe complications €42 664.

    Reply: In the abstract we have mentioned the study period and p-values were added.

    Changed in Manuscript:
    ‘A retrospective cohort analysis was performed using the Dutch Pancreatic Cancer Audit (DPCA) database of patients who underwent a PD at our institute between 2013 and 2017.’

    ‘Median costs for an uncomplicated course were €25 682 and for a complicated course €32 958 (p=0.001). The median costs for minor complications were €30 316 and for major complications €42 664 (p=0.001).’

  2. In introduction: in the sentence: A pancreatoduodenectomy (PD) is a complex procedure usually performed in case 37 of malignant tumours of the pancreatic head or surrounding structures: "surrounding structures" should be precised.

    Reply: In the introduction the ‘surrounding structures’ were specified.

Changed in Manuscript:
‘A pancreatoduodenectomy (PD) is a complex procedure usually performed in case of malignant tumours of the pancreatic head or surrounding structures like the duodenum, the ampulla of Vater or the distal bile ducts.’

  1. In material an methods: in the sentence: All consecutive adult patients who underwent PD (classic Whipple 66 procedure/pylorus resecting pancreatoduodenectomy (PRPD) or pylorus preserving pancreatoduodenectomy (PPPD)) between 2013 and 2017 at our institute were included in the study, the number of patients should be presented.

    Reply: In the material and methods the number of patients were presented.

    Changed in Manuscript:
    ‘This study is a retrospective cohort analysis using the DPCA database of patients who underwent a pancreatoduodenectomy at our tertiary referral center, the University Medical Centre Groningen (UMCG), between 2013 and 2017. In total 160 patients were included´

  2. In Table 1, there are 3 unplanned readmissions with no complications. What were the reasons of these readmissions?

Reply: The 3 unplanned readmissions with no complications were patients that were admitted for observation due to malaise, pain or nausea complaints and were discharged rapidly after complaints disappeared.

  1. In Table 1, regarding pathology, "benign" and "other" should be precised.

    Reply: In table 1 regarding pathology ‘benign’ was further specified
  2. In Table 1, regarding origin of tumor, and "other" should be precised. Why the tumor origin was unknown in 10 patients?

    Reply: In table 1 regarding origin of tumor. Other/unknown means that data was missing in our database or it was scored as other in the database.

  3. In Tables 2-4, types of statistical tests and p values should be presented.

    Reply: In table 2-4 the types of statistical test and p-values were added (see appendix section).

  4. Regarding Figure 1. A. Median total hospital costs (Uncomplicated course and complicated course by Clavien-Dindo classification); B. Median total hospital costs (Complicated course by CCI score), statistical difference between these groups should be analyzed. The post-hoc tests for finding statistical difference should be used.

    Reply: In the description of figure 1A and 1B p-values were added. Figure 1A and 1B are based on the results of table 2b and 3 so the results are based on the statistical tests used in table 2b and 3.

  5. The general characteristics (including age, gender, BMI, comorbidities, tumor location and histopathology including TNM classification, preoperative biliary stenting, type of PD, vacular resection (if it was performed), ASA, duration of hospitalization) should be compared between complication vs. non-complications and readmission vs. non-readmission groups.

    Reply: Preoperative biliary stenting, type of PD, vascular resection and duration of hospitalization were added to the baseline characteristics table for complication vs. non-complications and readmission vs. no-readmission as well as all types of complications. The variable TNM classification was unfortunately not included in our database.

    Changed in Manuscript: In the results in the baseline characteristics section the follow text was added:
    A total of 125 (78.1%) of the patients underwent a PPPD and 35 (21.9%) patients underwent a Classic Whipple procedure. A vascular resection was performed in 26 patients (16.3%). In total 97 patients (62.9%) underwent preoperative biliary stenting. In 80 patients (50%) the tumor originated from the pancreas. In 120 patients (70%) the pathology showed an adenocarcinoma.

  1. In results, all types complications should be presented (such as pancreatic fistula type A, B, C, wound infection, bile leak, DGE, etc. ). Also, correlation between types of complications and costs would be interesting for readers.

    Reply: All types of complications were added to the baseline characteristics table 1a and 1b.

    Changed in Manuscript: In the results in the baseline characteristics section the follow text was added:
    ‘In total 12 patients (7.5%) developed a clinically relevant (grade B/C) postoperative pancreatic fistula (CR-POPF). (Table 1a and 1b baseline characteristics).’

  2. What about length of hospital stay? Was a difference between patients with and without complications or according to Clavien-Dindo classification or complication types? The association betwen duration of hospitalization and complications types and classifications as well as duration of hospitalization and costs should be analyzed.

    Reply: Length of hospital stay was also added to table 1b, 2a,2b, 3 and 4.

Changed in Manuscript: In the results in the complicated course section the follow text was added:
‘The mean and median days of hospital stay were respectively 11.7 days and 11 days for patients with an uncomplicated course after PD. For patients with a complicated course (in general) after PD the mean and median days of hospital days were respectively 23.3 days and 17 days (p=0.001). The mean and median days of hospital stay for patients with minor complcations were respectively 17.1 days and 15 days. For patients major complications the mean and median days of hospital days were respectively 37.6 days and 31 days (p=0.001). (Table 2a)’

Changed in Manuscript: In the results in the CCI score course section the follow text was added:
‘The mean and median days of hospital stay for patients with a CCI score between 0-30 was ranging respectively from 12.4 - 19 days and 12 - 18 days. The mean and median days of hospital stay for patients with a CCI score between 30-50 was ranging respectively from 28.4 – 34.6 days and 26 – 31 days. In case of a CCI score of greater than 50, the mean and median days of hospital stay were respectively 58.7 and 53 days. Hospital stay seems to longer when the CCI score increases (p=0.001). (Table 3)’

Changed in Manuscript: In the results in the unplanned readmissions section the follow text was added: ‘The mean and median days of hospital stay were respectively 17.1 days and 13 days for patients with no readmissions. For patients with readmissions the mean and median days of hospital days were respectively 31.7 days and 26 days (p=0.001). (Table 4a)’

  1. What was the impact of comorbidities on complications and costs? It should be analyzed.

    Reply: We think we have provided a detailed and precise cost overview for complications by Clavien-Dindo classification as well as the CCI score and readmissions and by doing so we have answered our research question/study goals. The impact of comorbidities on complications and costs is another path that exceeds our current study goal and research question. Furthermore an extensive multivariate analysis on comorbidities and costs was not possible due to the limited time. It would also mean that our whole study design and article would have needed a major overhaul and would have to be changed from its core.

  2. Additional, advanced statistical methods (regression analyses) should be used in this study. Cost ratio for the increase in costs related to complications should be calculated using regression analysis. Multivariate linear regression was used to deter-mine The association between specific complications and costs should be calculated using multivariate linear regression analysis.

    Reply: Unfortunately due to limited time it was not possible to do an extensive regression analysis. A regression analysis would also mean that our whole study design and article would have needed a major overhaul. In consultation with our statistician we decided not to perform a regression analysis but to look at the costs by Clavien-Dindo classification and CCI score. We think we have provided a detailed and precise cost overview for complications by Clavien-Dindo classification as well as the CCI score and readmissions and by doing so we have answered our research question/study goals.

  3. Limitations of the study are presented. What about strengths? The authors have not explained what was the novelty of their study. The presented analyses are interesting and precise, but it is generally known that complications are associated with higher hospitalization costs because of longer hospital stay and additional conservative or surgical treatment.

    Reply: The strengths of the study were described in the discussion section. The strength of our study is that we have a larger study population (160 patients) and a more detailed hospital costs analysis compared to previous studies.

Reviewer 2 Report

The paper has been improved in this revised form

Author Response

1. The paper has been improved in this revised form.

Reply: Thank you very much. We also think that the comments and suggestions of the reviewers have improved the overall manuscript significantly.

Round 3

Reviewer 1 Report

Generally, it is an interesting article and the authors have changed their manuscript considering most my suggestions. A lot of additional informations have been presented. There are still some issues.

Major issue:

I understand that some of additional analyses have been not performed due to the limited time, but the multivariate linear regression analysis for predictors of hospital costs should be performed. Currently, only basic univariate tests are presented.

Minor issue:

The p-values are very important in presented comparisions and they should be presented in tables including in the main manuscript, not appendix.